# Mechanism Study on the Effect of Surface Electrical Property on Microbial Membrane Formation Efficiency of TiO_2_-SiC Composite Filler in Recirculating Aquaculture System

**DOI:** 10.3390/ma17143501

**Published:** 2024-07-15

**Authors:** Jiaxin Li, Ze Hong, Jingying Ouyang, Han Zheng, Ying Liu

**Affiliations:** 1School of Marine Science, Technology and Environment, Dalian Ocean University, Dalian 116023, China; 18341122909@163.com (J.L.); 13262614992@163.com (Z.H.); 18742521716@163.com (J.O.); 2Key Laboratory of Environment Controlled Aquaculture (Dalian Ocean University) Ministry of Education, Dalian 116023, China

**Keywords:** recirculating aquaculture system, biofilm culturing efficiency, titanium dioxide thin film, positive electro-adsorption

## Abstract

Recirculating aquaculture systems (RASs) offer significant advantages in aquaculture by markedly decreasing water usage and increasing culture density. A vital component within a RAS is the filler material, which serves as a surface for microbial colonization. Effective microbial treatment is crucial for the efficient operation of a RAS as it assists in purifying the wastewater generated within the system. Nevertheless, traditional fillers often show low efficiency in biofilm formation. The commercial silicon carbide used in this study is a foam ceramic filter with a density of about 0.4–0.55 g/cm^3^, a number of holes of about 10, and a through porosity of 80.9%, with a diameter of about 5 cm. This research investigates the utilization of a titanium dioxide–silicon carbide (TiO_2_-SiC) composite filler to improve the purification efficiency of ammonia nitrogen and chemical oxygen demand (COD) in aquaculture wastewater. The study involved the application of titanium dioxide films onto the surface of silicon carbide to produce the composite filler. This method takes advantage of the dipole interaction between titanium dioxide and microorganisms, which enhances biofilm culturing efficiency on the silicon carbide surface. The performance of three different fillers was assessed for their ability to purify aquaculture wastewater. Results showed that the TiO_2_-SiC composite filler was 1.67 times more effective in removing COD and 1.07 times more effective in removing ammonia nitrogen compared to using silicon carbide alone. These results demonstrate that the incorporation of a titanium dioxide coating substantially boosts the microbial colonization efficiency of silicon carbide, thereby enhancing the overall wastewater purification efficiency in RAS.

## 1. Introduction

Aquaculture is becoming an increasingly essential source of food for both current and future populations. As a result, ensuring the rapid and safe expansion of aquaculture as a dependable protein source is a primary objective for China at this juncture. Presently, China leads the world in both aquaculture area and total production. However, the rapid growth of this industry has exposed its environmental susceptibilities. Issues such as the depletion of fishery resources and the degradation of the marine environment have surfaced, primarily due to the unregulated discharge of aquaculture wastewater. These environmental challenges are a cause for concern [1,2].

A recirculating aquaculture system (RAS) eliminates residual feed, waste products, and toxic substances such as NH_3_-N and NO_2_-N from aquaculture water by utilizing mechanical filtration and biological treatment methods. Afterward, the treated water undergoes disinfection, oxygenation, removal of impurities, and temperature adjustment before being recirculated back into the aquaculture environment. This process facilitates the reuse of water in aquaculture operations [3,4]. Due to these distinct advantages, RAS has garnered increasing attention and has been extensively studied and applied in recent years.

The application of fillers in recirculating aquaculture systems (RAS) is a critical technology for the efficient purification of aquaculture wastewater, significantly influencing the effectiveness of microbial biofilm formation and the smooth operation of water circulation [5,6]. Fillers provide surfaces for biofilm attachment, which are predominantly composed of nitrifying bacteria, denitrifying bacteria, ammonia-oxidizing bacteria, water-purifying bacteria, and a few protozoa. These biofilms form a complex micro-ecosystem with specific structural and functional attributes [7]. Biological treatment technology, especially biofilm methods, plays an indispensable role in the purification of aquaculture wastewater and is the most widely used and cost-effective method for seawater purification in China [8,9]. The surface properties of fillers, including specific surface area, roughness, and electrical characteristics, are crucial as they directly impact microbial colonization and the longevity of microorganisms on the filler surfaces. Consequently, the development and optimization of filler materials are of immense value for enhancing the performance of RAS [10,11,12]. For instance, Dezhao Liu and colleagues [13] designed a polybutylene succinate/bamboo powder (PBS/BP) composite to act as both a solid carbon source and a biofilm substrate in a denitrification biofilter for RAS wastewater treatment. Their findings indicated that the PBS/BP composite exhibited a high potential for nitrate removal, making it a promising material for future research. Similarly, Guozhi Luo et al. [14] employed polycaprolactone (PCL) as a filler to enhance nitrate removal efficiency in RAS. The study revealed that PCL supports the growth of all three major types of denitrifying bacteria, demonstrating its effectiveness in improving water quality. Furthermore, Yulei Zhang and his team [15] developed a novel filler composed of 90–91% polyethylene, 3–4% PVA, and 2–3% polyvinylpyrrolidone (PVP) with a specific gravity of 0.4051 g/cm³. This PVA-PVP filler showed excellent performance in the removal of both ammonia and nitrate nitrogen, indicating its potential as a biofilm filler in RAS applications. These studies underscore the importance of selecting and developing appropriate fillers to enhance the efficiency of wastewater purification in recirculating aquaculture systems.

Despite their widespread use, traditional fillers often suffer from low biofilm formation efficiency in recirculating mariculture systems. The filler used in this experiment was silicon carbide, chosen for its low cost, simplicity, and unique pore structure. However, silicon carbide’s adsorption effect on microorganisms was initially inadequate, but this was improved in this study. To enhance adsorption, the experimental silicon carbide was coated with a titanium dioxide thin film, leveraging the dipole effect between the titanium dioxide and microorganisms for better microbial absorption. The commercial silicon carbide fillers have a specific surface area of approximately 15–20 m^2^/g. TiO_2_, a widely utilized semiconductor oxide, is notable for its roles in organic pollutant degradation, clean energy production, and self-cleaning due to its chemical stability, non-toxicity, and cost-effectiveness [16,17,18,19]. Among its various applications, wastewater treatment is particularly significant. Titanium dioxide possesses properties ideal for filler use, including chemical stability, a positively charged surface, hydrophilicity, and abundant availability [20]. These characteristics make TiO_2_ an excellent choice for filler material. Similarly, silicon carbide offers advantages such as a large specific surface area, stable physicochemical properties, and high-temperature resistance [21], enabling it to endure the annealing temperatures necessary for forming titanium dioxide films. Consequently, the combination of TiO_2_ and SiC presents a promising solution for improving membrane hanging efficiency in recirculating aquaculture systems.

In our study, we prepared TiO_2_-SiC composite fillers with varying layers of titanium dioxide thin film using the tension membrane method to control the number of layers. After loading the titanium dioxide thin film onto the silicon carbide filler, we subjected it to the microbial membrane hanging process under the same conditions as those used for single silicon carbide fillers. The TiO_2_-SiC composite filler exhibited a dipole effect due to the positive charge on the titanium dioxide surface, which interacts with negatively charged microorganisms. This interaction facilitates better adhesion of microorganisms to the filler surface, reducing the likelihood of detachment and enhancing the efficiency of microbial film formation. Additionally, the porous structure of the TiO_2_-SiC composite filler improves contact between microorganisms and the wastewater, thereby increasing the purification efficiency for pollutants in aquaculture wastewater. The large size of the silicon carbide filler also supports its recyclability [22,23]. This study aimed to produce TiO_2_-SiC composite fillers using the tension membrane method and to facilitate microbial adhesion on the filler surface through a natural membrane hanging method. We specifically tested the efficiency of microbial membrane formation and the effectiveness in purifying ammonia and COD in aquaculture wastewater for both TiO_2_-SiC and single silicon carbide fillers. Furthermore, we characterized the titanium dioxide using X-ray diffraction and assessed its surface electrical properties.

## 2. Experimental Details

### 2.1. Materials

The main experimental instruments used in this experiment include a particle size analyzer (Bettersize2600, Dandong Baite Instrument Co., Ltd., Dandong, China), a fluorescence microscope (XSP-63XDV, Shanghai Optical Instrument Factory 1, Shanghai, China), a muffle furnace (SX2-10-12N, Shanghai Yiheng, Shanghai, China), an oven (DHG-9070A, Shanghai Yiheng), an ultrasonic vibration meter, and other equipment. The reagents utilized in this experiment included acetylacetone (analytical grade), anhydrous ethanol (99%), anhydrous glucose (analytical grade), concentrated hydrochloric acid (35%), tetrabutyl titanate hydrolysis (analytical grade), standard ammonium acetate (0.01 mol/L), sodium hydroxide solution (25%), sulfuric acid solution (1:4 volume ratio), sodium carbonate solution (0.5 mol/L), sodium thiosulfate solution (0.01 mol/L), potassium permanganate solution (0.01 mol/L), starch solution (5 g/L), potassium iodide solution (0.5 g/mL), sodium hypobromite solution (0.01 mol/L), and naphthyl ethylenediamine dihydrochloride (1.0 g/L).

### 2.2. Preparation of TiO_2_-SiC Composite Filler

Firstly, 1 mL of acetylacetone was added to 35 mL of anhydrous ethanol (99%) and stirred to obtain a mixed solution. Then, 10 mL of tetra butyl titanate hydrolysis was gradually introduced into this mixture to produce mixture A. In parallel, 2 mL of ultrapure water and 0.5 mL of concentrated hydrochloric acid (35%) were mixed with 10 mL of anhydrous ethanol (99%) to create mixture B. Mixture A was then combined with mixture B and stirred for 15 min. The resulting solution was left at room temperature for 24 h, forming a bright yellow titanium dioxide precursor solution [24,25]. Eighteen pieces of silicon carbide filler were cleaned by ultrasonic washing with ethanol and ultrapure water and then dried. Six of these fillers were immersed in the titanium dioxide sol-gel for 30 s and dried at 80 °C for 90 s, resulting in TiO_2_-SiC composite fillers with a single layer of titanium dioxide film, labeled as group b. Another six fillers underwent the same process but with three repetitions of the titanium dioxide film coating, producing TiO_2_-SiC composite fillers with three layers of titanium dioxide film, labeled as group c. The remaining six silicon carbide fillers, which did not receive any titanium dioxide film, served as the control group, labeled as group a. All three groups were calcined and annealed at 500 °C in a muffle furnace for 60 min [26]. This process resulted in two groups of TiO_2_-SiC composite fillers and one control group.

The calculation formula for total bacterial number [27]:(1)$$E=\bar{x}\frac{S_{1}}{S_{2}}\frac{1}{V}$$

In the formula, *E* is the number of bacteria in the sample in units of bacteria per milliliter (bacteria/mL);

$\bar{\chi }$—the average value of the number of bacteria in each counting field;

*S*_1_—the area of the filter membrane in units of square millimeters (mm^2^);

*S*_2_—the area of the field of view of the oil microscope in units of square millimeters (mm^2^);

*V*—the volume of the filtered water sample in units of milliliters (mL).

### 2.3. Determination of Chemical Oxygen Demand

Briefly, sodium hydroxide solution (25%) and potassium permanganate solution (0.01 mol/L) were added to conical flasks and then dried at 130 °C for 30 min. After cooling the flasks to room temperature, sulfuric acid solution was added. The mixture was then combined with potassium iodide solution (1:4 volume radio), shaken until the solution turned yellow, and left in the dark for 5 min. Starch solution was added to the water sample, followed by titration with sodium thiosulfate solution until the blue color just disappeared. The volume, *V*, of sodium thiosulfate consumed by each water sample was recorded [28].
(2)$$\mathrm{COD}=\frac{c(V_{2}-V_{1})\times 8.0}{V}\times 1000$$
where *c* represents the concentration of sodium thiosulfate solution, mol/L; *V*_2_ denotes the volume of sodium thiosulfate solution consumed by titration when analyzing ultrapure water, mL; *V*_1_ represents the volume of sodium thiosulfate during titration of the sample, mL; *V* signifies the volume of the water sample, mL; and COD indicates the chemical oxygen demand of the water sample, mg/L.
(3)$$k=\frac{\ln(C/C_{0})}{t}$$
where *k* represents the decrease rate of COD value in the process of membrane hanging; *C* stands for COD values for each measurement, mg/L; *C*_0_ signifies the initial COD value, mg/L; and *t* denotes the days of membrane hanging.

### 2.4. Determination of Ammonia Nitrogen

Ammonium standard solution was added to 50 mL volumetric flasks, and then ammonia-free water was added to the calibration line. Ammonia nitrogen was measured using the hypobromite oxidation method. Absorbance values were recorded at a wavelength of 543 nm using 1 cm measuring cells, with ammonia-free distilled water serving as the reference. These absorbance values were used to calculate the ammonia nitrogen concentration in the sample solution based on the standard curve. Subsequently, the standard curve was plotted.

### 2.5. Simulation of Aquaculture Wastewater Purification Experiment

In our work, we loaded 2 g/L of titanium dioxide onto silicon carbide to form the titanium dioxide thin film by the tension membrane method. The weight ratio of titanium dioxide to silicon carbide is as follows: the weight ratio of silicon carbide composite fillers loaded with one layer of titanium dioxide film is about 1%, while the weight ratio of silicon carbide composite fillers loaded with three layers of titanium dioxide film is about 1.5%. Three plastic beakers were filled with coarse-filtered seawater to simulate the purification of aquaculture wastewater. Based on a carbon-to-nitrogen ratio of 6:1, the required initial amounts of chemicals were calculated to be 3.443 g of ammonium chloride and 11.59 g of anhydrous glucose [20]. These chemicals were added to each beaker of seawater. The prepared TiO_2_-SiC composite fillers (groups b and c) and the single silicon carbide filler (group a) were fixed on plastic nets using thin ropes, allowing each group to be suspended in the culture solution, with aeration controlled for each beaker. An equal amount of pre-membrane hung filler was added to each beaker as a source of microbial colonies, with microorganisms being naturally incubated under the same conditions of temperature, pH, and aeration [21]. Subsequently, experiments were conducted simultaneously to simulate the purification of COD and ammonia nitrogen in aquaculture wastewater. The concentrations of COD and ammonia nitrogen were tracked using the alkaline potassium permanganate method and the sodium hypobromite method, respectively. The cycle of this experiment is 24 days, with water samples measured every three days and worked for two cycles.

### 2.6. Characterization of the Surface Potential and X-ray Diffraction

Zeta potential was measured using a DLS Zeta potential analyzer (ZS90) from Malvern Instruments (Malvern, UK). X-ray powder diffraction (XRD) patterns were recorded using a Rigaku (Tokyo, Japan) D/MAX-2500 X-ray diffractometer with Cu Kα radiation.

## 3. Results and Discussion

### 3.1. X-ray Diffraction (XRD) Characterization of Titanium Dioxide Thin Film

First, X-ray diffraction (XRD) characterization was performed on the thin film prepared from the titanium dioxide precursor. As shown in Figure 1, the diffraction peaks at 25.22°, 37.78°, 47.94°, 54.96°, 62.69°, 69.76°, and 75.21° correspond to the {101}, {004}, {200}, {211}, {204}, {012}, and {002} crystal planes of the anatase phase of titanium dioxide, respectively (JCPDS No. 21-1271). The two characteristic peaks at 2θ = 27.41° and 42.08° are attributed to the {110} and {011} crystal planes of the rutile phase of titanium dioxide, respectively (JCPDS No. 21-1276). The absence of other characteristic peaks in the XRD pattern indicates that no other crystal phases were formed during the preparation process. Thus, the films deposited on the surface of the silicon carbide filler by the tension membrane method consist of titanium dioxide with both anatase and rutile phases.

### 3.2. Surface Electrical Potential Test of Titanium Dioxide Film at Different pH Values

The surface electrical potential of the titanium dioxide film and single SiC were tested using a particle size analyzer to verify that the surface was indeed positively charged. As shown in Figure 2, the potential value of the titanium dioxide thin film was approximately 20 mV when the test water environment was neutral (pH = 7), confirming the positive charge. However, the surface electrical properties of a single silicon carbide filler were positive under pH 6–8. With a pH of around 6.5, the surface electrical properties of a single silicon carbide filler were positive in the environment, with values around 6.4 mV. When pH is 7.2, the surface electrical properties of a single silicon carbide were around 5.3 mV. With a pH of around 8.0, the potential value of the single silicon carbide filler was approximately 4 mV. However, the intended application environment for the TiO_2_-SiC composite filler is seawater, which is weakly alkaline due to the hydrolysis of weak acidic anions, with a pH around 8. Therefore, the water environment was adjusted to test the surface electrical property under weakly alkaline conditions, simulating seawater pH. The data indicated that the titanium dioxide thin film retained a positive charge in this environment, with values around 10 mV. Further testing in a weakly acidic environment revealed that the surface electrical property of the titanium dioxide thin film was inversely proportional to the pH of the water environment. As pH increased, the surface electrical properties decreased and showed an inverse relationship, but still being positively charged. The surface electrical properties of single silicon carbide fillers are much lower than those of TiO_2_-SiC composite fillers. The specific surface areas of the composite fillers of the silicon carbide loaded with three titanium dioxide films, the composite fillers of silicon carbide loaded with one titanium dioxide film, and the single silicon carbide fillers are 19 m^2^/g, 17 m^2^/g, and 16 m^2^/g, respectively. The specific surface area has almost no effect on the degradation of ammonia nitrogen and COD, as we used commercial silicon carbide materials in our study, with little difference in specific surface area between 15 and 20 m^2^/g. These results confirm that the titanium dioxide film maintains a positive charge in a weakly alkaline environment, providing a theoretical foundation for enhancing the coupling effect between titanium dioxide and microorganisms to improve the purification efficiency of aquaculture wastewater on the surface of the silicon carbide filler.

### 3.3. Total Bacteria Number Record by Fluorescence Microscope Direct Count Method

In our study, the total bacterial count on the surface of three fillers was determined using the fluorescence microscope direct count method as specified in GB17378.7-2007 [29]. Table 1 presents the total bacterial count recorded using this method. Group a data represents the bacterial count on single silicon carbide carriers, group b data corresponds to TiO_2_-SiC carriers with a single layer of titanium dioxide thin film, and group c data reflects TiO_2_-SiC composite carriers with three layers of titanium dioxide thin film. According to the microscopic measurements, the total bacterial counts for the three groups were calculated using Formula (2). The results indicated that the total bacterial count was highest in group c, followed by group b, and lowest in group a. This suggests that the surface of the silicon carbide carriers coated with titanium dioxide thin film supports a higher bacterial load compared to the uncoated silicon carbide carriers. Among the groups, group c, with three layers of titanium dioxide thin film, exhibited the highest bacterial count. This is likely due to the positively charged surface of the titanium dioxide film, which enhances bacterial attachment. Consequently, the bacterial counts for groups c and b were significantly higher than for group a. Additionally, during the annealing process, the presence of three layers of titanium dioxide precursor in group c resulted in a larger film area, increased film thickness, and better bacterial adsorption performance. As a result, the total bacterial count observed in group b was less than that in group c, with group c showing the highest bacterial count. Therefore, it can be inferred that TiO_2_-SiC composite carriers with three layers of titanium dioxide thin film have the strongest adsorption capacity for microorganisms (bacteria) and exhibit the best biofilm formation among the three groups of silicon carbide carriers.

### 3.4. Chemical Oxygen Demand (COD) Removal Rate

COD, which measures the amount of oxidant consumed to treat a water sample under specified conditions, is commonly used to quantify the organic matter in the water sample [30,31]. The histogram data in Figure 3 indicate that all four groups of TiO_2_-SiC composite fillers significantly reduced COD values in mariculture wastewater during microbial biofilm formation, but the COD content of the single TiO_2_ group had almost no change. The removal effect of the single TiO_2_ group was hardly obvious because TiO_2_ requires light to undergo photocatalysis, so in the dark state, TiO_2_ has no degradation effect on COD in RAS. Notably, the two groups of silicon carbide fillers coated with thin films of titanium dioxide demonstrated a faster rate of COD reduction in aquaculture wastewater compared to the single silicon carbide fillers and single TiO_2_ group. This enhanced performance is attributed to the positively charged surface of the titanium dioxide film, which facilitates the attachment and growth of negatively charged microorganisms (such as nitrifying bacteria) on the silicon carbide filler surface, thereby improving microbial biofilm formation efficiency. Moreover, the silicon carbide composite filler loaded with three layers of titanium dioxide exhibited a faster COD reduction rate in aquaculture wastewater than the filler with only one layer of titanium dioxide. This suggests that the increased film thickness and surface area provided by the additional layers of titanium dioxide enhance microbial adsorption and activity, leading to more efficient COD purification.

The COD was calculated based on the test data in Section 2.3 and Formula (2).

As indicated by the COD value reduction rate curves in Figure 4, the slopes of the curves for groups d, c, b, and a, calculated using Formula (3), were 0.0863, 0.07061, 0.05168, and 0.00078, respectively. This means that the rate of COD removal for the two groups of composite fillers coated with titanium dioxide thin films (groups c and d) was higher than that for the single silicon carbide filler (group b) and the single titanium dioxide (group a). The removal effect of single TiO_2_ on COD in RAS under dark conditions is not obvious, and the values of COD remain almost unchanged within 0–20 days. This is because TiO_2_ requires light to undergo photocatalysis, so in the dark state, TiO_2_ has no degradation effect on COD in RAS. Specifically, the TiO_2_-SiC composite filler with three layers of titanium dioxide film exhibited the fastest rate of COD removal in aquaculture wastewater (group d), being 1.67 times faster than the single silicon carbide filler and 1.22 times faster than the TiO_2_-SiC composite filler with one layer of titanium dioxide.

### 3.5. Ammonia Nitrogen Removal Rate

In the ammonia nitrogen purification experiment, a single silicon carbide filler and a single TiO_2_ were compared with the TiO_2_-SiC composite filler. Figure 5 illustrates the change in ammonia nitrogen concentration over a 20-day period for the control and experimental groups, calculated from the standard curve of ammonia nitrogen. Figure 5 shows that the initial ammonia nitrogen concentration was the same for all four groups of fillers. Throughout the experimental purification process, a significant decrease in ammonia nitrogen concentration was observed, except in the single TiO_2_ group. After 20 days, the ammonia nitrogen concentration in the aquaculture wastewater treated with the two groups of TiO_2_-SiC composite fillers was significantly lower than that in the wastewater treated with the single silicon carbide filler and single TiO_2_. The removal effect of single TiO_2_ on ammonia nitrogen in RAS under dark conditions is not obvious, and the values of ammonia nitrogen remain almost unchanged within 0–20 days. This is because TiO_2_ requires light to undergo photocatalysis, so in the dark state, TiO_2_ has no degradation effect on ammonia nitrogen in RAS. The removal ratio of ammonia nitrogen for the single silicon carbide filler was approximately 88%. For the TiO_2_-SiC composite filler with one layer of titanium dioxide thin film, the removal ratio exceeded 91%. For the TiO_2_-SiC composite filler with three layers of titanium dioxide thin film, the removal ratio exceeded 95%.

This improvement can be explained by the enhanced adhesion efficiency of the microbial biofilm on the titanium dioxide-coated surface of the silicon carbide filler, which, in turn, increases the ammonia nitrogen removal rate. After 20 days, the TiO_2_-SiC composite fillers demonstrated higher microbial biofilm formation efficiency compared to the single silicon carbide filler and the single TiO_2_, resulting in better ammonia nitrogen purification in the aquaculture wastewater. Additionally, the TiO_2_-SiC composite filler with three layers of titanium dioxide exhibited the highest ammonia nitrogen removal efficiency, which was 1.07 times greater than that of the single silicon carbide filler.

### 3.6. Membrane Hanging Effects of Three Kinds of TiO_2_-SiC Composite Fillers

Figure 6 illustrates the biofilm formation effects of three different TiO_2_-SiC composite fillers. Group a represents the control group of a silicon carbide filler without titanium dioxide film, group b consists of a TiO_2_-SiC composite filler with one layer of titanium dioxide film, and group c includes a TiO_2_-SiC composite filler with three layers of titanium dioxide film. Picture d shows TiO_2_ in a sol–gel state, which is used to load TiO_2_ on silicon carbide filler by film pulling method, and picture e shows single TiO_2_ after annealing. After annealing, sol–gel state TiO_2_ will turn into powder. The surface of the single silicon carbide filler in group a showed almost no microbial biofilm, as the individual microorganisms (bacteria) were too small to be visible to the naked eye, although the presence of bacteria could be detected through testing.

In contrast, microbial biofilms were observed on the surfaces of the TiO_2_-SiC composite fillers in groups b and c. Specifically, the area covered by the biofilm on the TiO_2_-SiC composite filler with three layers of titanium dioxide film (group c) was larger than that on the filler with one layer of titanium dioxide film (group b). The following reasons explain this observation: 1. The TiO_2_-SiC composite filler with titanium dioxide film has a stronger adsorption ability for microorganisms than the single silicon carbide filler, making it easier for a biofilm to form on the TiO_2_-SiC composite filler surface. 2. The biofilm-forming area of the three layers of titanium dioxide film is larger than that of the single layer on the silicon carbide filler surface, and the film is thicker, allowing more microorganisms to adhere to and disperse across the filler surface.

Therefore, the TiO_2_-SiC composite filler with three layers of titanium dioxide film covered a larger microbial biofilm area and demonstrated higher purification efficiency for aquaculture wastewater.

## 4. Conclusions

In this paper, three titania–silicon carbide composite fillers with varying numbers of titania layers were prepared using the tension membrane method to address the low purification efficiency of conventional fillers in recirculating aquaculture systems. The membrane formation and surface electrical properties of the prepared titanium dioxide films were characterized through XRD and zeta potential tests. Data tracking of the three different TiO_2_-SiC composite fillers in the purification of ammonia nitrogen and COD in aquaculture wastewater indicated that the addition of titanium dioxide thin film significantly enhanced the purification efficiency of silicon carbide fillers. Specifically, the TiO_2_-SiC composite filler with three layers of titanium dioxide film removed 1.67 times more COD and 1.07 times more ammonia nitrogen than the single silicon carbide filler. Therefore, the titanium dioxide film improved the overall purification efficiency of aquaculture wastewater when used with silicon carbide fillers. Additionally, the composite model combining oxide and conventional fillers shows excellent potential for enhancing the purification efficiency of aquaculture wastewater and broadening the selection of effective fillers.

## Figures and Tables

**Figure 1 materials-17-03501-f001:**
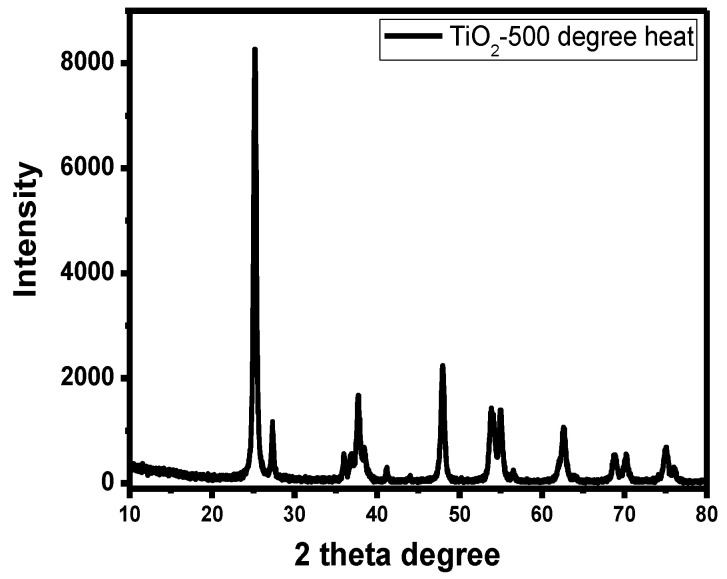
XRD characterization of titanium dioxide thin films.

**Figure 2 materials-17-03501-f002:**
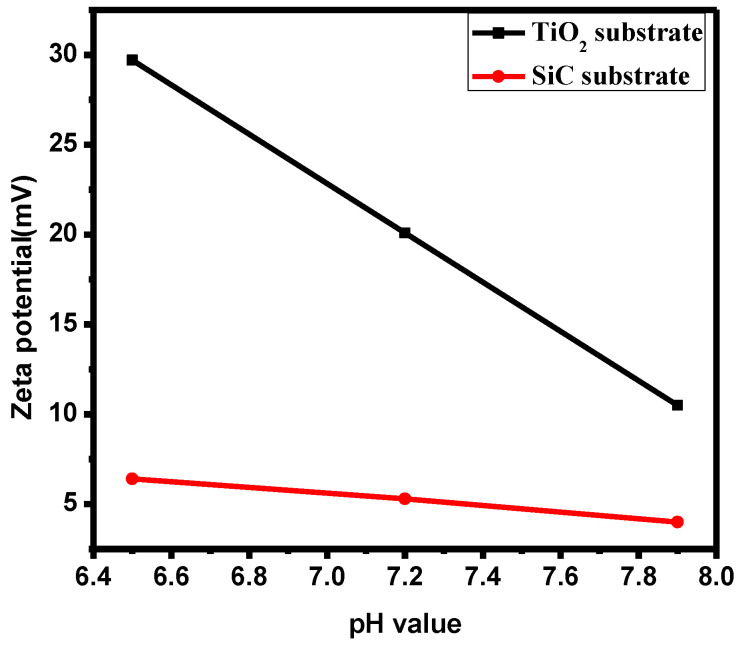
Surface electrical properties of titanium dioxide thin films and single SiCs at different pH values.

**Figure 3 materials-17-03501-f003:**
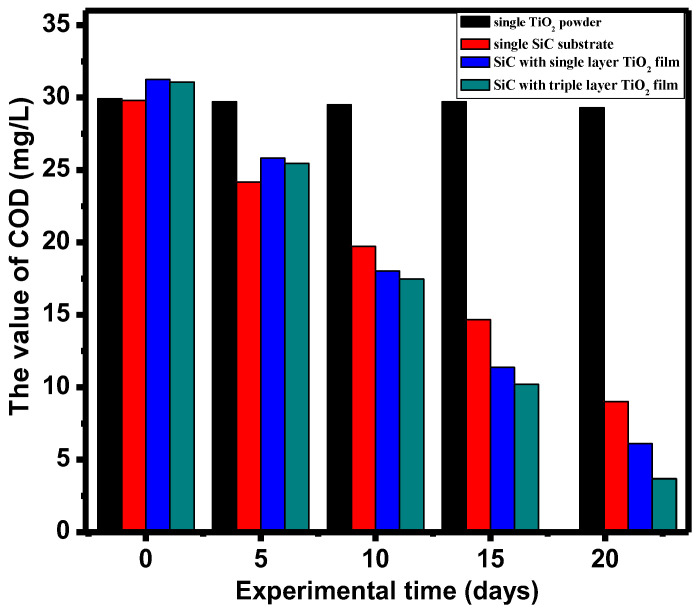
The removal rate of COD value in purifying aquaculture wastewater with four types of fillers.

**Figure 4 materials-17-03501-f004:**
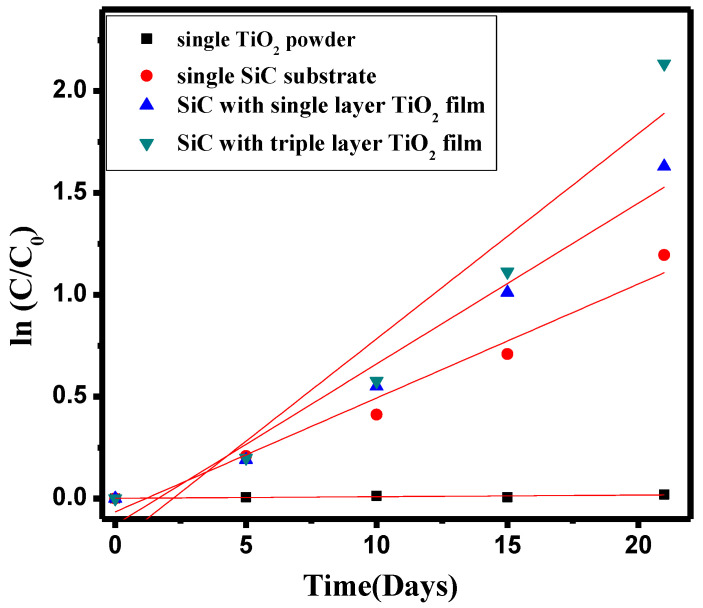
The rate of change of COD value in four types of fillers during film hanging.

**Figure 5 materials-17-03501-f005:**
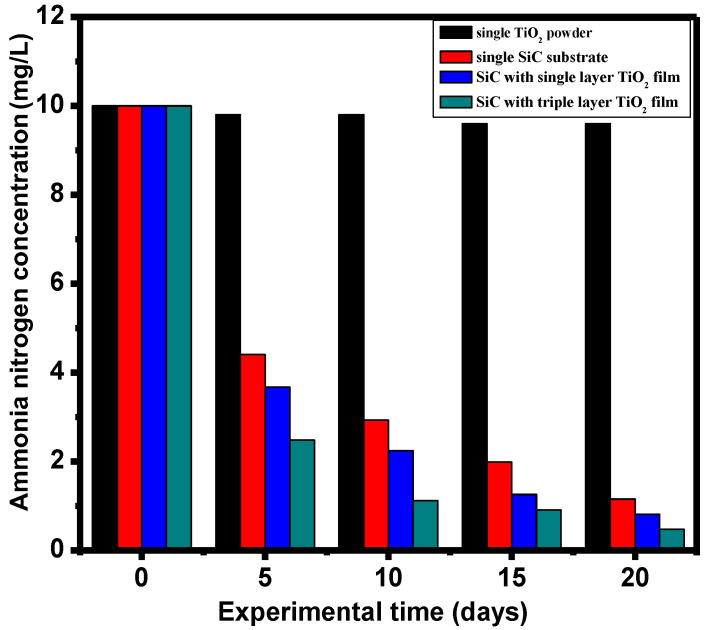
The removal rate of ammonia nitrogen concentration in purifying aquaculture wastewater with four types of fillers.

**Figure 6 materials-17-03501-f006:**
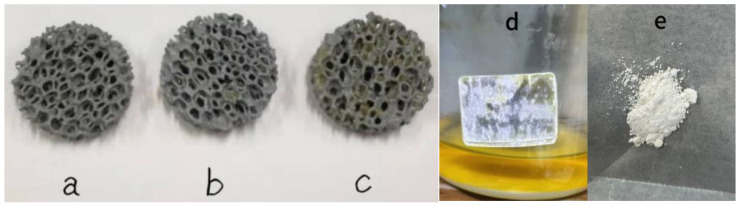
Result pictures of three kinds of TiO_2_/SiC composite fillers after 21 days of film hanging and state diagram of single TiO_2_ before and after annealing.

**Table 1 materials-17-03501-t001:** Total bacteria number record table.

Sample Number	Sample Amount/mL	Count Record					Mean	Bacteria Number/ (Bacteria/0.0001 mm^2^)
a	0.2	6	7	6	11	11	7	3452
		4	4	5	8	7		
		3	6	3	16	6		
		12	7	3	7	2		
b	0.2	12	20	23	25	15	15	7397
		16	15	17	11	16		
		10	9	11	6	20		
		15	14	19	16	5		
c	0.2	36	20	16	12	17	21	10,355
		14	8	18	15	20		
		39	6	13	5	11		
		15	21	17	14	106		

## Data Availability

The original contributions presented in the study are included in the article, further inquiries can be directed to the corresponding author.

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
