# Peer review of "Mechanism Study on the Effect of Surface Electrical Property on Microbial Membrane Formation Efficiency of TiO2-SiC Composite Filler in Recirculating Aquaculture System"

_materials, 2024, doi:10.3390/ma17143501_

Round 1
Reviewer 1 Report
Comments and Suggestions for Authors
Re: Mechanism Study on the Effect of Surface Electrical Property on Microbial Membrane Formation Efficiency of TiO2-SiC Composite Filler in Recirculating Aquaculture System.
Keyword: The key words are too long. They are like sentences.
The title for introduction is missing. This should be before line 29.
Page 38, you mentioned such as………..but you did not list anything. Kindly complete the statement.
“Recirculating aquaculture system (RAS) removes residual bait, excreta, and harmful substances such as ……………. from aquaculture waters through mechanical filtration and biological treatment technologies, and then transports them back to aquaculture water after disinfection, an increase of oxygen content, removal, and temperature regulation, so as to recycle aquaculture water”.
The sentence is too long, thus lost its meaning.
Line 129. Kindly state the concentration of NaOH and the potassium permanganate solution you used.
Line 131, was sulfuric acid concentrated? Or diluted, please state the concentration if it was diluted.
Line 131, what do you mean by configure potassium iodide? I would advise that you rewrite section 1.3 and provide detail information for your work to be reproducible.
Line 137, I would rewrite it like this “ A standard solution of ammonium were prepared by adding ammonia-free water to the ammonia standard (mention different concentrations prepared). The standards were used to calibrate the instrument (mention the name of the instrument, and manufacturer)”. “A standard curve was obtained after the calibration (include the R2 and the equation of the line here). The measurement were done at 543 nm wavelength and pathlength was 1 cm”.
Line 144 -147, should be moved to either introduction or discussion. “The filler utilized in this experiment was silicon carbide, which is inexpensive, simple, and convenient to take, and has a special pore structure. However, the adsorption effect of silicon carbide on microorganisms was not acceptable and improved in this experiment. – Move to appropriate place. This is method section.
Line 148, I would write that “titanium dioxide was loaded on silicon carbide” (please state their ratio).
I would advise that you rewrite your entire method sections. No section for the characterization methods. You need to describe how you did the XRD and mention the name of the instrument you used. How did you determine the surface electrical property?
All the formulas should be put in the method section.
You will need to do a statistical analysis of your results. How many times did you repeat each experiment?
Comments on the Quality of English LanguageThe author needs to improve the manuscript, because some sentences were not complete.
Reviewer 2 Report
Comments and Suggestions for Authors
The manuscript describes the use of three different fillers (probably better described as suspended growth media) on the removal of COD and ammonia from aquaculture wastewater. This is an interesting topic of commercial importance, but the manuscript needs to improve its presentation and analysis before publication. Below are comments to help improve the quality of the manuscript and these should be addressed.
1. Lines 72 to 77 describe the advantages of using titanium dioxide and silicon carbide as fillers and lists their large specific surface as an advantage. However, large specific surface area is not a material property but rather is a result of the materials form. Hence, you need to be more specific to identify what you are describing. Are you describing titanium dioxide or titanium dioxide nanoparticles? Please add in a description of the form of titanium dioxide and silicon carbide that you are describing or else remove the reference to surface areas.
Lines 109-127 describe the preparation of TiO2 and SiC fillers. The description should allow the work to be reproduced by others, but currently the description is not sufficiently detailed to allow this. For instance, the concentrations of chemical used is not included. Please add more detail to better describe your method.
3. Line 129-135 describes the technique to measure COD. An equation to convert the measured volume to COD is required to enble complete understanding of the technique.
4. Line 152 refers to "drug mass". I think you mean the required amount of chemicals. Drug mass is an unusual term and could be better described. Please review use of this term. Similarly line 154 refers to "drugs" but again this term should be reviewed and might be better described as chemicals. Drugs refer to chemicals that are used for a biological effect on the body or living organism.
5. Where the surface area of the fillers measured? This data would be good to include as it would enable the results to be presented in terms of colonies per m2 of filler. If the surfae areas could be measured that you benefit the manuscript.
6. Section 2.2 is titled "Surface electrical property ..". The surface property measured is the electrical potential of the surfaces, so the title would be better written as "Electrical potential of titanium dioxide ...." Similar changes could also be made elsewhere in the document e.g. title, etc.
7. Throughout section 2.2 there are results for the surface potential provided numerically and without any units eg. 20. However, the measured results have the units of mV and these should be added to the description in the text e.g. 20 mV rather than 20.
8. Where the surface potential and XRD characterisation performed for the SiC filler? These results are not included but are required for comparison against the TiO2 coating results.
9. Table 1 provides results for the concentration of bacteria /mL. However, given you are comparing the impact of fillers, it would be better to describe this as bacteria/surface area rather than bacteria per volume. Perhaps you could include both measurements.
10. Lines 211 to 218: Where the bacteria counted in solution or on the surface of the fillers or both? I think you are measuring bacteria on the surface of the fillers but it is unclear. Please be specific as to where the bacteria are that you are measuring as it will better clarify the results and their meaning for the readers.
11. Line 222: Please place a full stop (.) after "Group A".
12. Lines 222 to 234 describes there being more bacteria on the surface of the TiO2 coated fillers than on the SiC filler and attributes this to the surface potential characteristics. However, bacterial adhesion will also be impacted by the total surface area of the fillers and the surface roughness. Hence, having a measure of the surface area and roughness of the fillers before and following TiO2 coating would assist in determining if the extra bacteria on the TiO2 surfaces was a result of their being a larger surface area and possibly increased roughness or if it was only because of changes in surface potential. If the surface area and roughness measurements could be made and the analysis revised taking into account these parameters, it would greatly improve the robustness of the outcomes from the manuscript.
Comments on the Quality of English Language
The English expression is understandable but they use uncommon phrasing in places that can make the meaning hard to interpret.
Reviewer 3 Report
Comments and Suggestions for Authors
The authors conducted a mechanism study on the effect of surface electrical property on microbial membrane formation efficiency of TiO2-SiC composite filler in recirculating aquaculture system. I have some comments for the authors to consider:
Abstract
The abstract is difficult to comprehend in its current form. I suggest the authors revise the abstract to provide a consistent and clear understanding of the background, the problem they are trying to solve, how it was solved, their results, and conclusion.
Line 9, please provide a brief introduction about filler before going into their key roles.
Line 16, what does “specific methods” mean?
Introduction
Line 38, such as what?
Line 37-41, it is difficult to comprehend this sentence, please revise.
Section 1.3
What does “briefly” mean here? Generally, I recommend that authors carefully check this section and other sections and re-write them for clarity. What is the source of the water sample? Is this just an ultra-pure or is the water sample to be treated? No mention of how much of these reagents/solutions were used in each case.
Line 146, convenient to take or make?
Comments on the Quality of English LanguageModerate English check required.
Round 2
Reviewer 1 Report
Comments and Suggestions for Authors
I would advise that the author should do statistical analysis of the data. I know much research of this nature, did not do repeat the experiment, but is important to repeat the experiment at least three times and then find the average and standard deviation. This will help to correct errors that could be introduce due to instrument or human.
Also, figures 1 and 2 should contain information of both TiO2, SiC, and TiO2-SiC.
Also, the experiment needs to be conducted using TiO2 alone. Figure 3 - 6, have reported data related to SiC, and TiO2-SiC. Kindly provide data related to TiO2. We need to know what is responsible for the activity.
Comments on the Quality of English LanguageThe authors need to read their manuscript again and ensure they did not use long sentences.
Reviewer 2 Report
Comments and Suggestions for Authors
The authors have added surface area data to their results but have not used it in their analysis, and this should be done.
Comments on the Quality of English LanguageThe manuscript still requires improvement of the English expression and there was not mention of any improvement in this aspect of the manuscript in the response from the authors and there was no updated version of the paper for review.
Reviewer 3 Report
Comments and Suggestions for Authors
The abstract, introduction, and experimental details have improved significantly.
Minor comment:
Please define the acronym "RAS" where it was first mentioned in the abstract.
Comments on the Quality of English LanguageNo suggestion
Round 3
Reviewer 2 Report
Comments and Suggestions for Authors
My comments have been satisfactorily addressed.